# Urine Biomarkers in the Management of Adult Neurogenic Lower Urinary Tract Dysfunction: A Systematic Review

**DOI:** 10.3390/diagnostics13030468

**Published:** 2023-01-27

**Authors:** Periklis Koukourikis, Maria Papaioannou, Dimitrios Papanikolaou, Apostolos Apostolidis

**Affiliations:** 12nd Department of Urology, Aristotle University of Thessaloniki, General Hospital ‘Papageorgiou’, Thessaloniki 56429, Greece; 2Department of Biological Chemistry, Medical School, Aristotle University of Thessaloniki, Thessaloniki 54124, Greece

**Keywords:** neurogenic lower urinary tract dysfunction, neurogenic bladder, urine biomarkers

## Abstract

Background: Neurogenic lower urinary tract dysfunction requires lifelong surveillance and management for the perseveration of patients’ quality of life and the prevention of significant morbidity and mortality. Urine biomarkers are an attractive noninvasive method of surveillance for these patients. The aim of this systematic review is to search for and critically appraise studies that investigate the clinical usefulness of urine biomarkers in the management of neurogenic lower urinary tract dysfunction (NLUTD) in adults. Methods: This review was conducted according to PRISMA and MOOSE guidelines. Search strategy included PubMed, CENTRAL, and Scopus (until October 2022). Studies investigating potential urine biomarkers for the management of adults with NLUTD were included. Results: Fifteen studies fulfilled the criteria. To date, a variety of different urine molecules have been investigated for the diagnosis and management of neurogenic overactive bladder and detrusor overactivity (nerve growth factor, brain-derived neurotrophic factor, prostaglandin E2, prostaglandin F2α, transformation growth factor β-1, tissue inhibitor metalloproteinase-2, matrix metalloproteinase-2, substance P, microRNA), diagnosis of vesicoureteral reflux (exosomal vitronectin), urinary tract infection (neutrophil gelatinase-associated lipocalin, interleukin 6) and bladder cancer screening (cytology, BTA stat, survivin) in neurological patients. Conclusion: Further studies are needed to specify the utility of each molecule in the management algorithm of adult NLUTD.

## 1. Introduction

A biomarker is traditionally defined as any biological parameter that can be measured objectively and evaluated as an indicator of a physiological or pathological process or a response to a therapeutic intervention [1]. Biomarkers can be cells, genes, enzymes, proteins, etc. that can be detected in bodily fluids or tissues [2]. Urine is an important biological fluid in biomarker analysis as it can be obtained easily and non-invasively in large quantities and it contains several proteins, cells and biochemicals derived from the glomerular filtration of plasma, renal tubule excretion, and the urogenital system [3]. The importance of urine as a diagnostic tool was first noted by Hippocrates (460–355 BC); in *Aphorisms* he associated the bubbles on the surface of fresh urine with chronic kidney disease and linked the sediment of urine with fever [4]. In recent years, various urinary biomarkers have emerged that are used for the diagnosis and management of bladder and prostate cancer, kidney allograft rejection and overactive bladder (OAB) [5,6,7,8,9,10,11].

The lower urinary tract (LUT) consists of the urinary bladder and the bladder outlet (i.e., bladder neck, urethra, urethral sphincter and pelvic floor) and serves the functions of storage of urine at low pressures and periodic urine expulsion [12]. The activity of the two components of the LUT is coordinated by a complex neural system involving multiple areas at different levels from peripheral nerves in the pelvis to cortical areas in the brain [13]. As a result, a broad spectrum of neurological diseases and injuries of the brain and spinal cord, including multiple sclerosis (MS), cerebrovascular accident (CVA), spinal cord injury (SCI), spina bifida (SB) etc., can affect the neural control of the LUT causing neuro-urological symptoms. The combination of symptoms of OAB, including urinary urgency, frequency, nocturia and urgency incontinence, with voiding dysfunction and urinary retention is the most common clinical syndrome.

Adult neurogenic lower urinary tract dysfunction (NLUTD) according to the International Continence Society (ICS) refers to the abnormal or difficult function of the bladder and urethra (and/or prostate in men) in adults, in the context of a clinically confirmed relevant neurologic disorder [14]. NLUTD is associated with various complications including recurrent urinary tract infections (UTIs), incontinence, upper urinary tract deterioration, bladder stones and malignancy. Lifelong surveillance and management of NLUTD is mandatory to prevent these serious complications and preserve patients’ quality of life [15]. Surveillance strategies include laboratory tests (renal function, urinalysis), ultrasound, urodynamics and cystoscopy, although there is lack of consensus for the optimal long-term follow up [16,17]. Furthermore, urodymanics and cystoscopy are invasive, time-consuming, costly and a potential source of morbidity [18]. Hence, urine biomarkers are an attractive non-invasive diagnostic and surveillance tool in the management of patients with NLUTD.

Currently, the available studies on NLUTD and urine biomarkers have three main areas of research focus. The main research interest is the evaluation of the role of different molecules as potential biomarkers either in the diagnosis or in the assessment of the response to treatment and in the prediction of complications of neurogenic OAB [19,20,21,22,23,24,25,26,27]. Another area of research focus is the identification of a biomarker that can assist in differential diagnosis between asymptomatic bacteriuria and clinically significant urinary tract infections in patients with NLUTD [28,29]. Finally, other studies aim to elucidate the role of various biomarkers in the diagnosis of bladder cancer in this specific patient population [30,31,32,33].

This systematic review aims to summarize and critically appraise the existing evidence on the use of urine biomarkers in the management of adult NLUTD.

## 2. Material and Methods

This systematic review followed the reporting recommendations of the PRISMA statement and the guidelines of the meta-analysis of observational studies in epidemiology (MOOSE) [34]. We did not register a protocol for this systematic review. Moreover, this research did not receive any specific grant or funding. 

### 2.1. Search Strategy

We conducted a systematic literature search until October 2022 on three major electronic databases, including PubMed (Medline), CENTRAL (Cochrane Library) and Scopus looking for articles published in English without any publication date criteria. Additionally, so as to ensure literature saturation the reference lists from the included studies or reviews were hand-searched in order to retrieve additional studies that could have escaped the search. For the search strategy we used a combination of free text and medical subject headings (MeSH) of the terms “neurogenic bladder” or “neuropathic bladder” or “neurogenic lower urinary tract dysfunction” and “biomarkers” or “urine biomarkers” or “noninvasive biomarkers” or “cancer biomarkers”. After the Medline strategy was finalized, the search string was “translated” and used in other databases. The search was conducted independently by two authors (PK, DP) in a blinded manner. Any discrepancies were solved after consultation with a third investigator (AA). 

### 2.2. Eligibility Criteria

Retrospective or prospective studies or randomized controlled trials (RCTs) of adults who have been diagnosed with NLUTD, in whom urine biomarkers were assessed with the aim of identifying their potential use in the management of NLUTD, were eligible for this systematic review. Only English language studies were included. Case reports, animal studies and conference abstracts were excluded.

### 2.3. Study Selection Process

All relevant studies were imported into the Mendeley reference manager (Version 1.19 for Mac, Elsevier, Amsterdam, The Netherlands). Eligibility assessment was performed independently by three investigators (PK, DP, and AA) who screened the studies based on the predefined criteria. Disagreements were solved based on discussion. In the end, studies that were found to be eligible were screened as full text. 

### 2.4. Quality Assessment

A risk of bias assessment was performed by two reviewers (DP, and PK), as previously described in a blinded manner. As all the eligible studies were observational studies (prospective or retrospective cohort studies), they were evaluated using the Newcastle–Ottawa Scale (NOS) [35]. Based on a star system, each study was judged in three different categories, with three stars being the maximum score for each category. Studies were valued in the end as of high- (8–9 stars), medium- (6–7 stars) or low-quality (<6 stars). If any disagreement between the two reviewers was noted, it was arbitrated by a third one (AA). 

### 2.5. Data Extraction

Data from each study was extracted in an excel sheet that was designed a priori. The following details were collected: study design, biomarker evaluated, participants’ characteristics, and outcome analysis. Due to the great heterogeneity of the results, a meta-analysis was not possible. 

## 3. Results

The systematic search yielded 160 results following the removal of duplicates (Medline: 136, Cochrane: 11, Scopus: 54, manual searching: 6). After screening titles and abstracts, 128 were excluded as irrelevant. From the remaining 32 full-text studies, one could not be retrieved and 16 were excluded due to eligibility criteria. In the end, 15 studies were included in the qualitative synthesis and are illustrated in Table 1. Publication dates ranged between 1997 and 2021. Further details regarding the systematic search are provided in the PRISMA flow diagram (Figure 1) [36].

### 3.1. Study Characteristics

Yokoyama et al. [19]: a prospective single-center study investigating urinary nerve growth factor (uNGF) levels in various OAB patients. The study included 16 patients with NLUTD, all of them suffering from detrusor overactivity (DO), 10 due to SCI and six due to cerebrovascular diseases (CVD), and 32 controls. The uNGF levels were found significantly higher in patients with DO due to SCI compared to cystometrically normal controls (55.0 pg/mL vs. 12.2 pg/mL *p* = 0.0001). No difference was found between patients with DO due to CVD and normal controls (7.9 pg/mL vs. 12.2 pg/mL, *p* > 0.05).

Liu et al. [20]: a prospective single-center cross-sectional study evaluating the uNGF levels in 243 patients with proven DO with or without treatment with antimuscarinics, after treatment with detrusor botulinum toxin A (BoNT-A) injections and 38 controls. The NLUTD population consisted of 59 patients with untreated DO, 16 with well-treated DO and 25 who failed treatment. Patients with untreated DO had significantly higher levels of uNGF/Cr compared to controls (0.62 pg/mL vs. 0.005 pg/mL, *p* = 0.000). Levels of uNGF/Cr were significantly higher in patients with high voiding detrusor pressure (>40 cm H_2_O) compared to patients with low voiding detrusor pressure (0.97 pg/mL vs. 0.32 pg/mL, *p* = 0.039). A significant decrease in uNGF/Cr levels was observed after therapy in responders (*n* = 14) to BoNT-A injections (0.43 pg/mL vs. 0.096 pg/mL, *p* = 0.033) but not in non-responders (*n* = 5) to BoNT-A injections (0.76 pg/mL vs. 1.64 pg/mL *p* = 0.129) at 3 months.

Liu et al. [21]: a prospective single-center study assessing uNGF levels in patients with CVA (*n* = 93) and controls (*n* = 40). The median uNGF/Cr level was significantly higher in all patients with CVA than in controls (0.13 pg/mL vs. not detectable), with wide variations in uNGF/Cr levels. No significant difference in uNGF/Cr levels was found between patients with different urological conditions (urgency urinary incontinence (UI), impaired awareness of UI, OAB or no OAB, *p* = 0.805) and with different urodynamic findings (DO, detrusor hyperactivity and impaired contractility, and detrusor underactivity, *p* = 0.321). The increasing level of the severity of neurological impairment according to the American Heart Association Stroke Outcome Classification was correlated with significantly higher uNGF/Cr levels (level C 1.53 pg/mL vs. level B 0.27 pg/mL *p* = 0.002; level C 1.53 pg/mL vs. level A not detectable, *p* < 0.001; level B vs. level A, *p* < 0.001).

Yamauchi et al. [22]: a prospective single-center study assessing the urinary levels of prostaglandin E_2_ (PGE_2_), prostaglandin F2α (PGF_2a_), NGF and substance *p* in patients with suprapontine brain diseases (*n* = 114) and controls (*n* = 27). The NLUTD cohort was classified into four groups: patients without LUT symptoms (*n* = 19); the increased bladder sensation group (IBS) (*n* = 37); the OAB-dry group (*n* = 22); and the OAB-wet group (*n* = 36). Urinary PGE_2_ levels were significantly higher in all four groups of patients compared with controls. Patients in the OAB-dry and OAB-wet groups had significantly increased levels of urinary PGE_2_ compared to the patient group without LUTS (*p* < 0.004 and *p* < 0.015, respectively). In terms of urinary PGF_2a_ levels, patients in the OAB-wet group had significantly higher levels compared to patients without LUTS (*p* < 0.001). Urinary levels of NGF and substance P were not significantly different between the four groups of patients and between NLUTD patients and controls. 

Krebs et al. [23]: a prospective single-center study comparing uNGF levels between patients with NLUTD due to SCI (*n* = 37) and controls (*n* = 10). The uNGF concentration was <10 pg/mL (minimal detection limit) in all investigated samples.

Richard et al. [24]: a single-center prospective study comparing the urinary levels of NGF, brain-derived neurotrophic factor (BDNF), PGE_2_, transforming growth factor beta-1 (TGFβ-1) and the tissue inhibitor of metallopeptidase 2 (TIMP-2) between NLUTD patients of different neurogenic etiology (multiple sclerosis *n* = 6, SCI *n* = 20, spina bifida *n* = 15). Patients suffering from multiple sclerosis had significantly higher levels of urinary NGF/Cr and BDNF/Cr compared to the other two groups (8 pg/mL vs. 0.56 pg/mL vs. 1.25 pg/mg; *p* = 0.001; and 88.3 pg/mL vs. 5 pg/mL vs. 4.8 pg/mg; *p* < 0.0001; respectively). Patients suffering from SCI had significantly lower urinary levels of TGFβ-1 compared to spina bifida patients (10.8 pg/mL vs. 39 pg/mg; *p* = 0.04). Urodynamic parameters were not significantly associated with any of the urine biomarkers levels. 

Peyronnet et al. [25]: a prospective single-center cohort study evaluating the predictive value of NGF, PGE_2_, BDNF, matrix metalloproteinase 2 (MMP-2), TIMP-2 and TGF-β1 for adverse urodynamic parameters and for upper urinary tract damage in NLUTD patients due to spina bifida (*n* = 40). In terms of urodynamic findings, the urinary levels of TIMP-2 and MMP-2 were significantly associated with poor bladder compliance (<20 mL/cmH2O) (OR = 18.3; *p* = 0.043 and OR = 12.1; *p* = 0.039, respectively). TIMP-2 was significantly associated with upper urinary tract damage upon contrast-enhanced computed tomography scanning (OR = 19.81; *p* = 0.02).

Li et al. [26]: a prospective multicenter study examining the role of urine exosome proteins in the prediction of vesicoureteral reflux (VUR) in NLUTD patients due to SCI (*n* = 60). The levels of the urinary exosomal vitronectin were significantly higher in VUR patients (*n* = 25) than in no-VUR patients (*n* = 35) (*p* < 0.0001), the sensitivity and the specificity of the biomarker was 80% and 82.9%, respectively. 

Von Siebenthal et al. [27]: a prospective single-center study assessing the urinary microRNAs in patients with benign prostatic obstruction-induced LUT dysfunction (BLUTD) (*n* = 16), NLUTD (*n* = 12) and controls (*n* = 12). The levels of urinary microRNAs miR-10a-5p and miR-363-3p were significantly higher in the NLUTD group compared to BLUTD and controls (*p* = 9 × 10^−5^ and *p* = 0.00022, respectively). Urinary microRNA miR-301b-3p levels were significantly higher in BLUTD patients compared to NLUTD patients and controls (*p* = 0.0071).

Forster et al. [28]: A secondary analysis of urine samples, collected from a cohort of NLUTD patients (*n* = 27) participating in a prospective clinical trial, assessing the correlation of urine neutrophil gelatinase-associated lipocalin levels (uNGAL) and likelihood of UTI. Patients of the likely-UTI group had significantly higher levels of uNGAL compared to the patients in the no-UTI group and the patients in the unlikely-UTI group (187 ng/mL vs. 37 ng/mL, *p* < 0.01; 187 ng/mL vs. 95 ng/mL, *p* < 0.05, respectively). A difference in uNGAL levels was also found between the no-UTI and unlikely-UTI groups (*p* < 0.01).

Sunden et al. [29]: a longitudinal cohort study evaluating the urinary interleukin-6 (IL-6) levels during asymptomatic bacteriuria (ABU) and UTI, its association with UTI symptom severity and the diagnostic accuracy of urine IL-6, IL-8 and neutrophils in the differential diagnosis between UTI and ABU in NLUTD patients due to lower motor neuron (*n* = 12) and spinal lesions (*n* = 11). Urine levels of IL-6 were found to be significantly increased during UTI in pooled and intra-individual comparisons to ABU (*p* = 0.0371 and *p* = 0.0021, respectively). A threshold value of 25 ng/L in the urinary levels of IL-6 had a specificity of 93% and sensitivity of 77% in the diagnosis of UTI episodes with high-scoring symptoms.

Stonehill et al. [30]: a retrospective single-center review assessing the use of urine cytology as an adjunct for the diagnosis of BCa in NLUTD patients suffering from SCI. All patients (*n* = 208) had undergone a bladder biopsy due to a long-term indwelling catheter, gross or persistent microscopic hematuria, bladder stone, progressive urethral stricture or suspicious cytology. The sensitivity and specificity of the suspicious cytology results for BCa diagnosis were 71% and 97%, respectively. 

Hess et al. [31]: a retrospective single-center analysis of patients with NLUTD caused by SCI, who were diagnosed with bladder cancer (*n* = 16). Eight from the eleven available cytologies were positive for bladder cancer diagnosis.

Davies et al. [32]: a prospective single-center cohort study evaluating the efficacy of BTA stat, cytology, and urinary survivin levels for the surveillance of bladder cancer (BCa) in SCI patients (*n* = 457). Three patients were diagnosed with non-invasive BCa during the study period, all three tests were negative before BCa diagnosis.

Pannek et al. [33]: a retrospective single-center study investigating the clinical usefulness of urine cytology in the detection of BCa in an NLUTD population (*n* = 79). The sensitivity and specificity of cytology for BCa diagnosis were 83.3% and 43.7%, respectively.

### 3.2. Methodologic Quality Assessment

Quality assessment was conducted using the NOS, as mentioned previously. Four of them were high-quality studies, seven medium- and four low-quality. Results of the quality evaluation are illustrated on Table 2.

## 4. Discussion

This is the first systematic review to evaluate studies that investigate the use of urine biomarkers in patients diagnosed with NLUTD. The fact that urine biomarkers can be easily collected by non-invasive methods makes them attractive to be used not only as a tool for diagnosis but also for the surveillance of various conditions in this specific group of patients. 

A common trait of most patients with NLUTD is neurogenic DO causing OAB symptoms, such as urinary frequency, urgency and incontinence, and carrying the risk of upper urinary tract deterioration [37]. A plethora of studies investigated the role of different molecules, such as neurotrophins (NGF and BDNF), proteins involved in the degradation of the extracellular matrix (TIMP-2 and MMP-2), prostaglandins and mediators of inflammation (PGE_2_, PGF2a, substance P and TGFβ-1) as potential biomarkers for the assessment of neurogenic DO.

NGF is undoubtedly the most studied urine biomarker in the NLUTD population. NGF is a small signaling protein that regulates the growth and survival of sensory and sympathetic fibers, which both have a vital role in the micturition pathways [38]. In the urinary tract, NGF is produced by the urothelium and detrusor smooth muscle cells [39]. In the non-neurogenic population, it was found that elevated levels of uNGF are associated with OAB [11,40,41,42]. However, there have been some conflicting results among the studies in NLUTD patients of various etiologies that were included in our review. More specifically, the initial study from Yokoyama et al. observed elevated uNGF/Cr levels in patients with neurogenic DO due to SCI compared to healthy controls [19]. Furthermore, Liu et al. showed that the increased levels of uNGF found in untreated patients with neurogenic DO were diminished after successful treatment with antimuscarinics and in the responders to intravesical BoNT-A injections. Positive correlations between uNGF/Cr levels and the urodynamic parameters of high voiding detrusor pressure and detrusor–sphincter dyssynergia were also observed, suggesting that uNGF can be a useful biomarker in the assessment of disease progression and treatment outcomes [20]. In contrast, Krebs et al. did not detect NGF in urine in a small cohort of patients suffering from SCI, raising questions about the validity of the ELISA kit produced by Promega Corporation, which was used in most of the previous studies [23]. Non-specific binding of this test antibody to the precursor of NGF has been related to false-positive results and low specificity, resulting in the withdrawal of the Promega test kit from the market in 2014 [43,44,45].

Concerning NLUTD due to suprapontine brain diseases, uNGF levels were not elevated in two studies [19,22]. In opposition, another study by Liu et al. did find raised uNGF in NLUTD patients due to CVA, even though the only positive correlation of the uNGF levels was with the severity of neurological impairment and not with the urological symptoms or the urodynamic parameters [21]. In this NLUTD group of patients, of the examined urine biomarkers (PGE_2_, PGF2a and substance P) only PGE_2_ correlated with the severity of OAB symptoms [22].

Two studies examined the association between multiple urine biomarkers (NGF, BDNF, PGE_2_, TGFβ-1, MMP-2 and TIMP-2) and urodynamic findings. In the study by Richard et al., none of the biomarkers tested were associated with urodynamic parameters. The levels of urine NGF/Cr and BDNF/Cr were associated with the etiology of NLUTD, and were found to be elevated in multiple sclerosis patients when compared to SCI and spina bifida patients [24]. These findings, along with the abovementioned findings in patients suffering from CVA, suggest that uNGF and uBDNF that are derived from the brain are excreted through the kidneys by plasma filtration [21]. In this regard, previous studies have shown elevated levels of BDNF in the serum and cerebrovascular fluid of patients suffering from ischemic stroke, traumatic brain injury and encephalitis [46,47,48]. 

Peyronnet et al. described positive associations between elevated urinary TIMP-2 and MMP-2 and poor bladder compliance (<20 mL/cmH_2_O) in patients with NLUTD due to spina bifida. This was the only study which assessed the diagnostic performance of the biomarkers; the area under the ROC curve (AUC) for the diagnosis of poor bladder compliance was 0.70 for MMP-2 and 0.69 for TIMP-2. In the same study, TIMP-2 was associated with upper urinary tract damage diagnosed by CT scans with an AUC of 0.72; a threshold of 405.9 pg/mL had sensitivity of 72.7% and specificity of 72.6% for diagnosing upper urinary tract damage [25]. In agreement, two studies that focused on a pediatric population suffering from NLUTD due to myelodysplasia demonstrated good diagnostic performance of urinary TIMP-2 for adverse urodynamic parameters, suggesting that invasive urodynamic studies in patients with low urinary TIMP-2 levels could be avoided during follow-up. More accurately, Sekerci et al. demonstrated that the TIMP-2 threshold of 6.95 ng/mg had an 80% sensitivity and 68.3% AUC in predicting detrusor leak point pressure (DLPP) > 40 cmH_2_O [49]. Nayak et al. showed an AUC of 0.977, 0.915, and 0.605 for predicting DLPP, compliance and percentage of expected bladder capacity, respectively [50]. These findings underscore the remarkable effect of extracellular matrix remodeling in bladder compliance, as the MMP family and their inhibitors (TIMPs) are the main enzymes involved in this process [51]. Furthermore, in a multicenter cohort of patients suffering from SCI, exosomal urinary vitronectin was found to be the only biomarker, among 18 exosomal urinary proteins, that could diagnose VUR with an 80% sensitivity and 82.9% specificity [26]. Vitronectin plays also a role in extracellular matrix remodeling and has been associated with various types of tissue fibrosis [52,53].

Urine miRNA is another biomarker that has been thoroughly investigated during the last two decades in the field of oncological urology. More specifically, in the field of prostate cancer there are currently commercially available urine tests evaluating the presence of various miRNAs, such as PCA3 (prostate cancer gene 3), HOXC6 and DLX1 miRNAs or TMPRSS2:ERG, which are used in the diagnosis or surveillance of men with prostate cancer [54,55,56,57,58]. However, in the field of bladder cancer, despite various available studies, a consensus has not been reached within the urologic society regarding the use of a specific biomarker in everyday practice [59,60,61]. Our systematic search has identified only one study that evaluates the use of miRNAs in patients with NLUTD [27]. The authors found that the urinary miRNA profile (using three specific miRNAs) could discriminate between healthy controls and patients suffering from BLUTD or NLUTD. Further studies are needed in order to validate these findings. However, their results have highlighted the differences noticed in the urine miRNAs’ composition between different age groups. It was previously shown that numerous age-related changes are noticed in the 24-h urine composition of patients with lithiasis [62]. In this regard, age matching of controls to the test subjects should be taken into account when designing future studies.

Another major issue for patients with NLUTD is the difficulty to accurately diagnose urinary tract infections (UTI) from asymptomatic bacteriuria (ABU). Impaired storage and voiding function results in elevated post-void residual urine that predisposes them to the risk of UTI [63]. Diagnosis is challenging as symptoms are often non-specific, thus UTIs in this group are associated with high morbidity and hospital utilization [64]. On the other hand, treating ABU with antibiotics may lead to bacterial resistance [63,65]. In this regard, Sunden et al. investigated whether urine IL-6 could help as an added tool to identify UTI [29]. Their study found that urine IL-6 concentration had a specificity of 93% and a sensitivity of 77% in diagnosing UTI when a threshold of 25 ng/L was used, and when urine WBC was added to the synthesis, sensitivity then reached 88%. Results are promising, however, due to the small sample size further studies are needed to validate these findings and possibly explore whether specific IL-6 concentrations could help stratify patients in groups with different needs for action. IL-6 has been investigated as a urine biomarker for bladder pain syndrome showing promising results as a tool in identifying Hunner’s ulcerative cystitis or other subtypes of bladder pain syndrome [66]. 

Another biomarker that has been investigated as a potential tool in diagnosing infection in this population is uNGAL. In the adult population it has been explored as a marker for identifying acute kidney injury, while in the paediatric population with NLUTD, it has been thoroughly studied as a tool that could aid in differentiating UTI from ABU [67,68,69,70,71]. In their study of adults with NLUTD, Forster et al. demonstrated that uNGAL can also be a promising marker which could help in differentiating UTI in this population [28]. However, this study suffered many limitations and further investigation is warranted to evaluate its potential role in diagnosing UTI. 

BCa in the NLUTD population is a major concern. Although its incidence is not as high as thought in previous years, the aggressiveness of the disease together with a younger age at diagnosis, higher percentage of squamous cell carcinoma and a higher mortality compared to the general population highlights the need for early diagnosis and treatment [72,73,74,75,76]. Screening cystoscopy and urine cytology for BCa in this specific population remains a matter of debate and there is a lack of recommendations in the clinical guidelines from the international societies [15,77]. 

Regarding urine cytology, three retrospective studies have reported a sensitivity ranging between 71 and 83.3% and two studies reported a specificity of 43.7–97% in SCI patients with long-term indwelling catheters or symptoms of BCa [30,31,33]. One prospective study in SCI patients showed that screening cytology, BTA tests and survivin were negative for malignancy in three patients diagnosed with BCa during the period of the study. Furthermore, 119 and 9 false positive results of BTA stat and survivin were detected, respectively, in patients without bladder malignancy [32]. BTA stat is a commercially available and easy to use test which detects human complement factor H-related protein produced by bladder cancer cells in urine [78]. The sensitivity of the BTA test outperforms urine cytology for BCa diagnosis in normal individuals but the specificity is lower [79]. Previous studies have also shown a high rate of false positive results in patients presenting with hematuria due to possible cross reaction of the test with circulating blood antigens [80]. In the SCI cohort, the presence of pyuria was associated with false positive results. These facts exclude this biomarker from the screening of NLUTD patients, as pyuria is prevalent and hematuria can result from chronic inflammation or micro-trauma during intermittent catheterization [32]. In a recent network meta-analysis, survivin, a protein member of the inhibitor of apoptosis proteins gene family, has shown the highest sensitivity, negative predictive value and accuracy among the other urine biomarkers for BCa diagnosis [81]. However, methodology errors when assessing the survivin levels in frozen-stored urine could have affected the results of the SCI cohort.

In our systematic review we tried to be as comprehensive as possible, adhering to the PRISMA and MOOSE guidelines. However, this study has some limitations. Most studies were of evidence level 2 or 3 (prospective cohort, controlled or not) and no randomized controlled trial compared the urine biomarkers to invasive methods of management. There was great heterogeneity regarding the biomarkers studied and the study designs, thus a meta-analysis was not possible. Furthermore, some of the studies were retrospective, which could possibly introduce confounding errors that could affect their results. Moreover, many of the included studies have a small sample size thus it is difficult to interpret their results with safety. Finally, our review did not include “grey literature”, thus some information related to our subject may have been missed.

## 5. Conclusions

This systematic review summarized the available evidence for urine biomarkers in the management of adults diagnosed with NLUTD. During the last two decades, numerous urine molecules have gained interest, which are involved in the diagnosis and surveillance of neurogenic overactive bladder and detrusor overactivity, the diagnosis of vesicoureteral reflux and urinary tract infection, and bladder cancer screening in this specific population. However, the paucity of evidence and conflicting results do not allow the application of any of the urine biomarkers in every day clinical practice. Further studies providing a high level of evidence are needed to specify the utility of each molecule in the management algorithm of adult NLUTD.

## Figures and Tables

**Figure 1 diagnostics-13-00468-f001:**
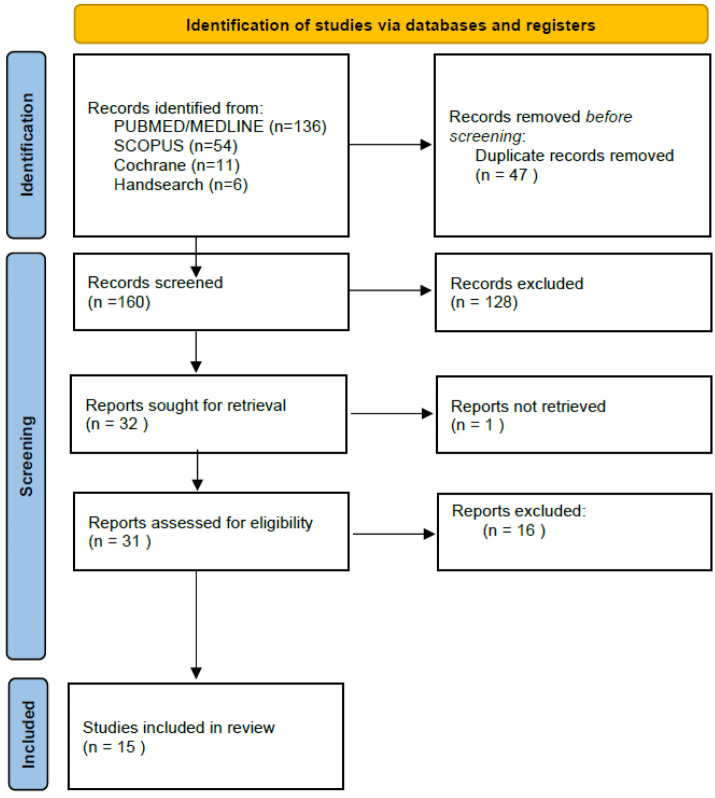
PRISMA 2020 flow diagram of study search and selection.

**Table 1 diagnostics-13-00468-t001:** Characteristics of the included studies.

Author	Type Of Study	Population	Patients (*N*)	Subgroups	Biomarker
Stonehill et al. [30]. (1997)	Single-Center Retrospective	NLUTD	208	SCI	Cytology
Hess et al. [31]. (2003)	Single-Center Retrospective	NLUTD	16	SCI	Cytology
Davies et al. [32]. (2005)	Single-Center Prospective	NLUTD	457	SCI	Cytology BTA Survivin
Yokoyama et al. [19]. (2008)	Single-Center Prospective	Mixed (1) NLUTD (2) nNLUTD (3) Healthy Controls	51 + 32	NLUTD (*n* = 16) - SCI (*n* = 10) - CVD (*n* = 6) nNLUTD (*n* = 35) - Idiopathic (*n*= 19) - BOO (*n* = 16) Controls (*n* = 32)	NGF
Liu et al. [20]. (2008)	Single-Center (Prospective) Cross-Sectional	Mixed	281	NLUTD (*n* = 100) nNLUTD (*n* = 143) Controls (*n* = 38)	NGF
Liu et al. [21]. (2009)	Single-Center Prospective	Mixed		NLUTD (*n* = 93) -CVA Controls (*n* = 40)	NGF
Yamauchi et al. [22]. (2010)	Single-Center Prospective	Mixed	141	NLUTD–SL (*n* = 114) Controls (*n* = 27)	PGE_2_, PGF_2a_ NGF Substance P
Krebs et al. [23]. (2016)	Single-center Prospective	Mixed	47	NLUTD -SCI (*n* = 37) Controls (*n* = 10)	NGF
Pannek et al. [33]. (2017)	Single-Center Retrospective	NLUTD	79	SCI (*n* = 65) MS (*n* = 8) Other (*n* = 6)	Cytology
Sunden et al. [29]. (2017)	Single-Center Prospective	NLUTD	23	DI (*n* = 12) SCI (*n* = 11)	IL-6, IL-8
Richard et al. [24]. (2019)	Single-Center Prospective	NLUTD	41	MS (*n* = 6) SCI (*n* = 20) SB (*n* = 15)	NGF. BDNF TGFβ-1, PGE_2_ TIMP-2
Peyronnet et al. [25]. (2019)	Single-Center Prospective	NLUTD	40	SB	NFG, BDNF, PGE2, MMP-2, TIMP-2, TGF-B1
Forster et al. [28]. (2020)	Multicenter Prospective	NLUTD	27	SCI (*n* = 24) SB (*n* = 1) MS (*n* = 2)	uNGAL
von Siebenthal et al. [27]. (2021)	Single-Center Prospective	Mixed (1) NLUTD (2) BLUTD (3) Healthy Controls	50	NLUTD (*n* = 12) BLUTD (*n*= 26) Controls (*n* = 12)	miRNA
Li et al. [26]. (2021)	Multicenter Prospective	NLUTD	60	SCI	Exosomal vitronectin

NLUTD: neurogenic lower urinary tract dysfunction, LUTD = lower urinary tract dysfunction, SCI = spinal cord injury MS = multiple sclerosis, CVD = cerebrovascular disease, BOO = bladder outlet obstruction, nNLUTD = non-neurogenic LUTD, CVA = cerebrovascular accident, DI = detrusor insufficiency, SB = spina bifida, BLUTD = benign prostatic obstruction LUTD, BTA = bladder tumor antigen, NGF = nerve growth factor, PGE_2_ = prostaglandin E_2_, PGF_2a_ = prostaglandin F2α, IL-6 = interleukin 6, IL-8 = interleukin 8, BDNF = brain-derived neurotrophic factor, TGFβ-1 = transformation growth factor β-1, TIMP-2 = tissue inhibitor metalloproteinase-2, MMP-2 = matrix metalloproteinase-2, uNGAL = urine neutrophil gelatinase-associated lipocalin, miRNA = microRNA.

**Table 2 diagnostics-13-00468-t002:** Quality assessment of the included studies using the Newcastle–Ottawa Scale.

Study	Case Definition	Representativeness of Cases	Selection of Controls	Definition of Controls	Cohort Representativeness	Selection of Non-Exposed Cohort	Ascertainment of Exposure	Outcomes of Interest Not Present	Sample Representativeness	Sample Size	Non-Respondents	Ascertainment of The Exposure	Matched Controls	Additional Factors	Ascertainment of Exposure	Ascertainment Of Participants	Non-Response Rate	Assessment of Outcome	Follow-Up Long Enough	Adequacy of Follow-Up	Assessment of The Outcome	Statistical Test	Total Score	Quality of Study
Stonehill et al. [30]. (1997)					*		*	*										*	*	*			6	Medium
Hess et al. [31]. (2003)					*		*	*										*	*	*			‘6	Medium
Davies et al. [32]. (2005)					*		*											*	*	*			5	Low
Yokoyama et al. [19]. (2008)	*	*		*									*	*	*	*							7	Medium
Liu et al. [20]. (2008)									*		*	*									**	*	6	Medium
Liu et al. [21]. (2009)					*	*	*	*					*	*				*	*	*			9	High
Yamauchi et al. [22]. (2010)	*	*	*	*									*					*	*	*			8	High
Krebs et al. [23]. (2016)					*	*	*	*					*					*	*	*			8	High
Pannek et al. [33]. (2017)					*		*	*										*					4	Low
Sunden et al. [29]. (2017)					*		*	*											*	*			5	Low
Richard et al. [24]. (2019)					*		*	*										*	*	*			6	Medium
Peyronnet et al. [25]. (2019)									*			*									**	*	5	Low
Forster et al. [28]. (2020)					*		*	*										*	*	*			6	Medium
von Siebenthal et al. [27]. (2021)	*	*		*									*	*				*	*	*			8	High
Li et al. [26]. (2021)					*		*	*										*	*	*			6	Medium

* Asterisks refer to the study characteristics used for the quality assessment according to the Newcastle–Ottawa Scale.

## Data Availability

Not applicable/No new data were created.

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
