# Peer review of "Urine Biomarkers in the Management of Adult Neurogenic Lower Urinary Tract Dysfunction: A Systematic Review"

_diagnostics, 2023, doi:10.3390/diagnostics13030468_

Round 1

Reviewer 1 Report

This systematic review summarizes 15 studies that fulfilled the criteria to search for the use of urine biomarkers in the diagnosis and surveillance of adult NLUTD. In the review, a variety of urine molecules have shown their correlation with NLUTD and correlated complications. However, the review also points out the limitations in most studies and the need for further investigation to evaluate clinical usefulness of these biomarkers.

Searching for better methods to improve the diagnosis and surveillance of NLUTD and its complications has remained important clinical aspects. Therefore, this review may provide a clearer direction for future research in this field, so as to explore the clinical value of urine biomarkers in the management of adult NLUTD through more convincing work.

There are some suggestions for the review:

(1) It may be better to briefly summarize the main focus of studies on NLUTD at present in the introduction, so as to show the importance or novelty of this topic.

(2) The definition about the NLUTD can be moved to the third paragraph in the introduction.

(3) When introducing NLUTD in the introduction, it is recommended to mention the causes of NLUTD, such as SCI, CVA, CVD, that are involved in the following studies. This may be better to introduce the later studies and also correspond to the sentence ”a broad spectrum of neurological diseases and injuries” in Line 48.

(4) There is inconsistent use in the abbreviations in the text. For example, “PGE2” in Line 158 is different from “PGE2” specified in Line 154. Please correct the similar mistakes in the text.

(5) A study of using vitronectin as a biomarker for VUR was mentioned at the end of the sixth paragraph in the Discussion, but I wonder if the presentation of this example is not well connected with the previous content about MMP and TIMP, and also the effect of extracellular matrix remodeling in bladder compliance. It may be possible to consider adding more description to strengthen the relationship.

Author Response

Reviewer’s comments to authors and detailed item-per-item responses

The authors would like to thank the editors and the reviewers for their time and effort in reviewing this manuscript. All reviewers’ suggestions and recommendations have been taken under consideration.

Reviewer: 1

General comments

This systematic review summarizes 15 studies that fulfilled the criteria to search for the use of urine biomarkers in the diagnosis and surveillance of adult NLUTD. In the review, a variety of urine molecules have shown their correlation with NLUTD and correlated complications. However, the review also points out the limitations in most studies and the need for further investigation to evaluate clinical usefulness of these biomarkers.

Searching for better methods to improve the diagnosis and surveillance of NLUTD and its complications has remained important clinical aspects. Therefore, this review may provide a clearer direction for future research in this field, so as to explore the clinical value of urine biomarkers in the management of adult NLUTD through more convincing work.

There are some suggestions for the review:

Suggestion #1

It may be better to briefly summarize the main focus of studies on NLUTD at present in the introduction, so as to show the importance or novelty of this topic.

Response #1

Thank you for your suggestion. A new paragraph was added in the Introduction section.

Action #1

Addition of a new paragraph in the Introduction section:

“Currently, the available studies on NLUTD and urine biomarkers have three main areas of research focus. The main research interest is the evaluation of the role of different molecules as potential biomarkers either in the diagnosis or in the assessment of the response to treatment and in prediction of complications of neurogenic OAB (19–27). Another area of research focus is the identification of a biomarker that can assist in the differential diagnosis between asymptomatic bacteriuria and clinically significant urinary tract infection in patients with NLUTD (28,29). Finally, other studies aim to elucidate the role of various biomarkers in the diagnosis of bladder cancer in this specific patient population (30–33).”

Suggestion #2

The definition about the NLUTD can be moved to the third paragraph in the introduction.

Response #2

Thank you for your comment.

Action#2

The NLUTD definition moved to the third paragraph of the introduction.

Suggestion #3

When introducing NLUTD in the introduction, it is recommended to mention the causes of NLUTD, such as SCI, CVA, CVD, that are involved in the following studies. This may be better to introduce the later studies and also correspond to the sentence ”a broad spectrum of neurological diseases and injuries” in Line 48.

Response #3

Thank you for your comment. Causes of NLUTD were added in the introduction.

Action #3

Addition of causes of NLUTD in the second paragraph of the Introduction section:

“As a result, a broad spectrum of neurological diseases and injuries of the brain and spinal cord, including multiple sclerosis (MS), cerebrovascular accident (CVA), spinal cord injury (SCI), spina bifida (SB) etc, can affect the neural control of the LUT causing neuro-urological symptoms”.

Suggestion #4

There is inconsistent use in the abbreviations in the text. For example, “PGE2” in Line 158 is different from “PGE2” specified in Line 154. Please correct the similar mistakes in the text.

Response#4

Thank you for your observation. The spelling errors were corrected.

Action#4

PGE2 was corrected to PGE2 in all text.

Suggestion #5

A study of using vitronectin as a biomarker for VUR was mentioned at the end of the sixth paragraph in the Discussion, but I wonder if the presentation of this example is not well connected with the previous content about MMP and TIMP, and also the effect of extracellular matrix remodeling in bladder compliance. It may be possible to consider adding more description to strengthen the relationship.

Response #5

Thank you for your suggestion. The end of the sixth paragraph in the discussion was revised.

Action #5

Revision of the end of paragraph 6 in the Discussion section:

“Furthermore, the diagnostic performance of 18 exosomal urinary proteins in the diagnosis of VUR was tested in a multicenter cohort of patients suffering from SCI, with exosomal urinary vitronectin found to be the only biomarker, among 18 exosomal urinary pro-teins, that could diagnose VUR with 80% sensitivity and 82.9% specificity (29). Vitronectin plays also a role in extracellular matrix remodeling and has been associated with various types of tissue fibrosis (52,53).”

Reviewer 2 Report

This is the first systematic review to summarize the available evidence for urine biomarkers in the management of adults diagnosed with neurogenic lower urinary tract dysfunction (NLUTD). Urine biomarkers can be easily collected by non-invasive methods, and therefore may represent a tool not only for diagnosis but also for surveillance of various conditions in these patients. This work provides detailed data about a variety of urine biomarkers investigated for the diagnosis and management of neurogenic overactive bladder and detrusor overactivity, diagnosis of vesicoureteral reflux, urinary tract infection and bladder cancer screening in this specific population. However, the authors highlight how  there is a paucity of evidence and conflicting results to allow the application of any of the urine biomarkers in every day clinical practice. Therefore, this review could represent a starting point for further studies of high level of evidence to specify the utility of each molecule in the management algorithm of adult NLUTD.

Concluding, I think that this manuscript is well written and needs only minor adjustments, in particular a punctuation and spelling check to eliminate typos.

Author Response

Reviewer’s comments to authors and detailed item-per-item responses

The authors would like to thank the reviewer for their time and effort in reviewing this manuscript. All reviewer’s suggestions and recommendations have been taken under consideration.

Reviewer: 2

General comments

This is the first systematic review to summarize the available evidence for urine biomarkers in the management of adults diagnosed with neurogenic lower urinary tract dysfunction (NLUTD). Urine biomarkers can be easily collected by non-invasive methods, and therefore may represent a tool not only for diagnosis but also for surveillance of various conditions in these patients. This work provides detailed data about a variety of urine biomarkers investigated for the diagnosis and management of neurogenic overactive bladder and detrusor overactivity, diagnosis of vesicoureteral reflux, urinary tract infection and bladder cancer screening in this specific population. However, the authors highlight how there is a paucity of evidence and conflicting results to allow the application of any of the urine biomarkers in every day clinical practice. Therefore, this review could represent a starting point for further studies of high level of evidence to specify the utility of each molecule in the management algorithm of adult NLUTD.

Concluding, I think that this manuscript is well written and needs only minor adjustments, in particular a punctuation and spelling check to eliminate typos.

Response

Thank you for your encouraging and rewarding comments. We have made extensive editing with corrections of grammar, punctuation and spelling errors throughout the manuscript.

Actions